# Management of Barrett’s Esophagus: Practice-Oriented Answers to Clinical Questions

**DOI:** 10.3390/cancers15071928

**Published:** 2023-03-23

**Authors:** Rocco Maurizio Zagari, Veronica Iascone, Lorenzo Fuccio, Alba Panarese, Leonardo Frazzoni

**Affiliations:** 1Department of Digestive Diseases, IRCCS Azienda Ospedaliero, Universitaria di Bologna, 40138 Bologna, Italy; 2Department of Medical and Surgical Sciences, University of Bologna, 40138 Bologna, Italy; 3Division of Gastroenterology and Digestive Endoscopy, Department of Medical Sciences, Central Hospital, Azienda Ospedaliera, 74100 Taranto, Italy

**Keywords:** Barrett, esophagus, diagnosis, therapy

## Abstract

**Simple Summary:**

The management of patients with Barrett’s esophagus still poses several clinical issues to the clinician, from correctly defining diagnosis to choosing adequate treatment. This brief and evidence-based review is aimed at providing a practical guide for the adequate management of this condition.

**Abstract:**

Barrett’s esophagus is the most important complication of gastro-esophageal reflux disease and the only known precursor of esophageal adenocarcinoma. The diagnosis and treatment of Barrett’s esophagus are clinically challenging as it requires a high level of knowledge and competence in upper gastrointestinal endoscopy. For instance, endoscopists should know when and how to perform biopsies when Barrett’s esophagus is suspected. Furthermore, the correct identification and treatment of dysplastic Barrett’s esophagus is crucial to prevent progression to cancer as well as it is the endoscopic surveillance of treated patients. Herein, we report practice-oriented answers to clinical questions that clinicians should be aware of when approaching patients with Barrett’s esophagus.

## 1. Introduction

Barrett’s esophagus (BE) is a condition characterised by the replacement of the normal squamous epithelium of the distal esophagus with specialized columnar epithelium with goblet cells [1]. This metaplastic change is caused by pathological gastroesophageal reflux and, is clinically relevant as it is the only known precursor of esophageal adenocarcinoma, a cancer with steadily increasing incidence in the last 40 years in Western countries [2]. The diagnosis and the management of BE are clinically challenging, as the gastroenterologist should know when to suspect BE, how to perform biopsies to confirm the diagnosis and eventually identify dysplasia, and how to treat and follow-up patients after endoscopic eradication of dysplasia or cancer. We report 12 practice-oriented answers to clinical questions that clinicians should be aware of, in order to improve the diagnosis and management of Barrett’s esophagus (Table 1).

Question 1.What is the prevalence of BE across Europe, the USA and Asia and which are the possible causes of this variability?Answer: The global prevalence of histologically confirmed BE in individuals with gastro-oesophageal reflux according to the Montreal definition [3] varies among countries, broadly resembling the geographical distribution of gastro-esophageal reflux itself. The highest prevalence is registered in North America (14.3%) and South America (12.5%), followed by Asia (6.1%), Europe (5.7%) and Middle-East (3.8%) [2]. This variability may be caused by different pathogenetic factors other than GERD itself and genetic characteristics [4]. First, the prevalence of *Helicobacter pylori* infection, which differs between the ethnic groups, has been found to be inversely associated with BE risk and may therefore act as a protective factor [5]. Second, obesity probably acts as an independent risk factor for BE, both through low levels of adiponectin and high levels of leptin that alter cell proliferation processes, and changes in the esophageal microbiota due to unhealthy dietary habits that promote carcinogenesis [6]. 

Question 2.Are there any biological biomarkers or genetic factors associated with BE and esophageal adenocarcinoma?Answer: BE is a known precursor of EAC, which has led to the recommendation that patients with BE undergo regular endoscopic surveillance. However, the diagnostic yield of endoscopy with the Seattle biopsy protocol has been questioned. Thus, in order to prevent the increasing morbidity and mortality from EAC, there is an urgent need for early detection and surveillance biomarker assays that are accurate, low-cost, and feasible to implement in clinical practice. Many biomarkers have been studied: altered TP53 tissue expression seems to be the most promising one [7,8], whereas p16, and higher circulating levels of leptin, glucose, insulin, CRP, IL6, and sTNFR-2 may be also associated with an increased risk of EAC [9]. More prospective studies are required to identify biomarkers that can help select high-risk individuals for targeted prevention and early detection.

Question 3.Should I perform an endoscopy in all patients with symptoms of gastroesophageal reflux disease to diagnose Barrett’s esophagus?Answer: Although gastroesophageal reflux disease (GERD) represents the pathophysiological primer for the development of BE, the prevalence of this condition in patients with GERD is very low, being about 2–5% in Europe [2]. Therefore, performing an upper gastrointestinal (GI) endoscopy to rule out BE in all patients with chronic symptoms of GERD is not feasible; it is necessary to stratify the risk of BE and prioritize endoscopy in patients with a higher probability of having this condition. International guidelines recommend performing upper GI endoscopy in patients with typical GERD symptoms and at least two of the following risk factors: age ≥50 years old, male sex, Caucasian ethnicity, obesity and first-degree familial history for BE or esophageal adenocarcinoma (EAC) [10,11].

Question 4.Why is it important to correctly identify diaphragmatic hiatus and esophagogastric and squamous-columnar junctions at upper GI endoscopy?Answer: At endoscopy, the diaphragmatic hiatus can be seen as a concentric narrowing of the lumen related to the diaphragmatic contraction during inspiration. The esophagogastric junction is defined by the top of the gastric folds, while the squamous-columnar junction (or Z line) represents the boundary between the columnar epithelium of the stomach and the squamous epithelium of the esophagus [12]. Correctly recognizing these landmarks during upper GI endoscopy is crucial for a correct diagnosis of hiatal hernia and BE. Hiatal hernia is defined when the esophago-gastric junction is proximally displaced ≥2 cm above the diaphragmatic hiatus, whereas BE occurs when the Z line is proximally displaced ≥1 cm above the esophagogastric junction. During endoscopy, it is important to avoid excessive insufflation of the lumen in order not to flatten gastric folds, which may cause a false impression of BE in a patient with a hiatal hernia. 

Question 5.Should I take biopsies of the proximally displaced Z line for less than 1 cm?Answer: A proximal displacement of columnar mucosa above the esophagogastric junction for less than 1 cm is defined as an irregular Z line. This finding is not uncommon, with a prevalence varying between 5% and 20%, in patients undergoing upper GI endoscopy [13]. When biopsies are performed, the irregular Z line generally reveals gastric or intestinal metaplasia. However, there is evidence that the malignancy risk of this condition, defined by the incidence of dysplasia or esophageal adenocarcinoma, is very small or almost null [13]; therefore, there is no indication to take biopsies of an irregular Z line or to follow-up patients with such a condition [11].

Question 6.Should I repeat endoscopy with biopsies in a patient with suspected Barrett’s esophagus if there is no intestinal metaplasia at histology?Answer: BE diagnosis is based on the presence of intestinal metaplasia, i.e., specialized columnar epithelium with goblet cells at histological examination. In a subgroup of patients with suspected BE, a histological examination may not reveal intestinal metaplasia due to an error in biopsy sampling. A cohort study of 80 patients with suspected BE, who were negative for intestinal metaplasia and underwent a repeat endoscopy with biopsies, reported that about 30% of them had intestinal metaplasia at the second endoscopy [14]. Therefore, international guidelines recommend repeating upper GI endoscopy at least once, possibly within 1–2 years, in patients with endoscopically suspected BE not confirmed at histology, before definitively ruling out this condition [1].

Question 7.Should Seattle biopsy protocol be systematically applied to Barrett’s esophagus surveillance?Answer: The Seattle protocol consists in taking four biopsies, i.e., one for each quadrant, at 2-cm intervals along the entire length of BE [15]. The application of this biopsy protocol increases the probability of detecting dysplasia in BE mucosa by reducing the sampling error, since areas of dysplasia may not be visible or have a focal and variable distribution, [15]. Unfortunately, the adherence of endoscopists to the Seattle protocol in clinical practice varies from 50% to 80% [15,16]. International guidelines recommend adhering to the Seattle protocol even when endoscopy is performed with advanced imaging techniques, such as conventional or virtual chromoendoscopy [17].

Question 8.Should I use chromoendoscopy in the surveillance of patients with Barrett’s esophagus?Answer: Conventional chromoendoscopy with dye and virtual chromoendoscopy increase the likelihood of identifying visible lesions in Barrett’s mucosa that may harbour dysplasia or adenocarcinoma. Chromoendoscopy with acetic acid involves the application of diluted acetic acid on Barrett’s mucosa, which initially assumes a whitish colour; such staining is lost more quickly from areas with dysplasia or adenocarcinoma (i.e., loss of aceto-whitening), thus allowing the performance of more targeted biopsies. A meta-analysis of nine studies confirmed that this technique has a sensitivity of 92% and a specificity of 96% for the diagnosis of high-grade dysplasia or esophageal adenocarcinoma [18]. In addition, technological evolution has led to endoscopes equipped with high-resolution optics which combined with digital filters of the spectrum of light allow the performance of virtual chromoendoscopy, such as narrow-band imaging (NBI). The use of NBI allows for the visualisation of the mucosal and vascular pattern of the esophagus, highlighting the potentially dysplastic or carcinomatous areas. A randomised trial including 123 patients with BE undergoing endoscopic surveillance compared NBI and high-definition white-light (HDWL) endoscopy for detecting dysplasia [19]. Dysplasia was more frequently identified in the NBI group than in the HDWL group (30% vs. 21%, *p* = 0.01). Therefore, guidelines recommend the routine use of chromoendoscopy with acetic acid or virtual chromoendoscopy in patients with BE under endoscopic surveillance, in addition to the Seattle biopsy protocol [11,17].

Question 9.When should I stop endoscopic surveillance in patients with Barrett’s esophagus?Answer: The expected benefits of BE surveillance decrease in parallel with the patient’s life expectancy. A recent cost-effective study [20] showed that the age threshold for stopping endoscopic surveillance depends primarily on gender and comorbidities. According to this study, endoscopic surveillance should be stopped at the age of 80 years in men and 75 years in women without comorbidities, and five years earlier in case of serious comorbidities. In accordance with this study, current European guidelines recommend discontinuing surveillance of non-dysplastic BE between 75 and 80 years of age [21]. Nevertheless, the health condition of patients and the economic status of different countries and regions vary greatly, therefore a tailored approach may be required.

Question 10.Should visible lesions harbouring dysplasia or adenocarcinoma be resected endoscopically before ablating Barrett’s esophagus?Answer: Performing endoscopic resection of visible lesions within BE before its ablation has two rationales. First, the sample provided for histological examination includes the *muscularis mucosae* and part of the submucosa and allows for properly staging the lesion when compared to biopsy; this can change the clinical management of patients in up to 40% of cases [22]. Second, performing ablation of BE after resection of visible lesions minimizes the likelihood of developing metachronous dysplasia or cancer in the remaining Barrett’s mucosa [11,17].

Question 11.Should patients with Barrett’s esophagus undergo anti-reflux surgery for anti-neoplastic purposes?Answer: Anti-reflux surgery can heal reflux esophagitis and improve reflux symptoms. Therefore, it would be logical to consider its application in BE patients to reduce the risk of developing dysplasia or adenocarcinoma. However, two considerations do not suggest recommending such a strategy. First, the risk of progression of nondysplastic BE to dysplasia or cancer is very low, being below 1% per year [17]. Second, there is no evidence that anti-reflux surgery, i.e. fundoplication protects against the development of dysplasia or cancer in BE patients. A study conducted on over 900,000 Northern-European patients with GERD, of which about 50,000 underwent anti-reflux surgery, did not show a reduction in the risk of EAC after anti-reflux surgery, even when considering only patients with severe esophagitis or BE [23]. Therefore, there is currently no indication of anti-reflux surgery for anti-neoplastic purposes in patients with BE.

Question 12.Should I discontinue endoscopic surveillance after endoscopic treatment of dysplastic BE or esophageal cancer and BE ablation?Answer: Patients with dysplastic BE or esophageal cancer who went through endoscopic ablation of nondysplastic BE after resection of visible lesions still have an increased risk of recurrence of BE, dysplasia and cancer. A meta-analysis including about 3000 patients who underwent endoscopic ablation of dysplastic BE estimated that the annual incidence of recurrence was 9.5% for nondysplastic BE, 2% for low-grade dysplasia and 1.2% for high-grade dysplasia or adenocarcinoma [24]. Therefore, patients who achieved confirmed endoscopic ablation of BE should be included in endoscopic surveillance programs [25]. During the examination, the distal esophagus should be carefully inspected, in particular immediately above the esophagogastric junction, with a high-resolution endoscope with chromoendoscopy to reveal any visible lesions, including subtle mucosal irregularities that may harbour sub-epithelial dysplasia, the so-called “buried dysplasia” [26].

## 2. Conclusions

Barrett’s esophagus is a condition that continues to be clinically relevant, being the only known precursor of esophageal adenocarcinoma, cancer that keeps rising in incidence in the Western world. This clinical review synthesizes current evidence-based best practices in the diagnosis and management of BE. Evolving knowledge and endoscopic technology are likely to improve the management of BE with a positive impact on the secondary prevention of esophageal adenocarcinoma in the next future.

## Figures and Tables

**Table 1 cancers-15-01928-t001:** Practice-oriented answers to clinical questions on the management of Barrett’s Esophagus.

**Question 1.** What is the prevalence of BE across Europe, the US, and Asia and which are the possible causes of this variability? **Answer:** BE prevalence broadly varies between countries, ranging from 3.8% in Asia to 14.3% in the US. This reflects the variability in GERD, *H. pylori* infection and obesity.
**Question 2.** Are there any biological biomarkers or genetic factors associated with BE and esophageal adenocarcinoma? **Answer:** Altered TP53 tissue expression, p16, and higher circulating levels of leptin, glucose, insulin, CRP, IL6, and sTNFR-2 have been found to be associated with progression to high-grade dysplasia and adenocarcinoma.
**Question 3.** Should I perform an endoscopy on all patients with symptoms of GERD to diagnose BE? **Answer:** Endoscopy should be reserved for patients with GERD symptoms and multiple risk factors for BE.
**Question 4.** Why is it important to recognize the diaphragmatic hiatus and the esophagogastric junction and squamous-columnar junctions at upper GI endoscopy? **Answer:** These are crucial landmarks for a correct endoscopic diagnosis of hiatal hernia and BE.
**Question 5.** Should I take biopsies of the proximally displaced Z line of less than 1 cm? **Answer:** Since the risk for malignancy of this condition is negligible, there is no indication to take biopsies.
**Question 6.** Should I repeat endoscopy with biopsies in a patient with suspected BE if the histological examination did not show intestinal metaplasia? **Answer:** Endoscopy with biopsies should be repeated at least once, possibly within 1–2 years.
**Question 7.** Should the Seattle biopsy protocol be systematically applied to BE surveillance? **Answer:** The Seattle protocol should be systematically applied to BE surveillance, even when using advanced imaging techniques.
**Question 8.** Should I use chromoendoscopy in the surveillance of patients with BE? **Answer:** Chromoendoscopy with dye, i.e. acetic acid, or virtual chromoendoscopy is recommended in the BE surveillance, in addition to the Seattle biopsy protocol, in order to increase the detection of dysplasia or cancer.
**Question 9.** When should I stop endoscopic surveillance in patients with BE? **Answer:** Endoscopic surveillance of BE should be stopped depending on patient age and comorbidities, approximately at 80 years of age in men and 75 years of age in women.
**Question 10.** Should visible lesions harbouring dysplasia or adenocarcinoma be resected endoscopically before ablating Barrett’s esophagus? **Answer:** Endoscopic resection of visible lesions with evidence of dysplasia or superficial adenocarcinoma should always be performed before ablation of the remaining BE mucosa for both clinical and prognostic purposes.
**Question 11.** Should BE patients undergo anti-reflux surgery for anti-neoplastic purposes? **Answer:** There is no indication of the need for anti-reflux surgery for anti-neoplastic purposes in patients with BE.
**Question 12.** Should I stop endoscopic surveillance after endoscopic treatment of dysplastic BE or esophageal cancer and BE ablation? **Answer:** Endoscopic surveillance should not be discontinued since there is an increased risk of recurrence of BE, dysplasia and cancer even after BE ablation.

BE: Barrett’s esophagus. GERD: gastroesophageal reflux disease.

## Data Availability

Not applicable.

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
