# Peer review of "Management of Barrett’s Esophagus: Practice-Oriented Answers to Clinical Questions"

_cancers, 2023, doi:10.3390/cancers15071928_

Round 1
Reviewer 1 Report
Answer to question 1 - please consider commenting on patients of other races (perhaps guidelines can be different, at least in the US at some clinics, it is so).
Has this opinion manuscript been invited ? The manuscript does not describe any novel answers, at least from the view of GI specialist. Nonetheless, if such opinion manuscript was invited then please consider answering on two additional questions:
1 - prevalence in the US is around 5.6%. comparison of rates to the rest of the world and the reasons of its variability?
2 - genetic factors/markers associated with BE and are there any genomic studies successfully validated it?
Author Response
Answer to question 1 - please consider commenting on patients of other races (perhaps guidelines can be different, at least in the US at some clinics, it is so).
Has this opinion manuscript been invited ? The manuscript does not describe any novel answers, at least from the view of GI specialist. Nonetheless, if such opinion manuscript was invited then please consider answering on two additional questions:
1 - prevalence in the US is around 5.6%. comparison of rates to the rest of the world and the reasons of its variability?
2 - genetic factors/markers associated with BE and are there any genomic studies successfully validated it?
RESPONSE: We thank the reviewer for the insightful comments. We have added two new Q&A paragraphs to our review (which are now no.1 and no.2), focusing on the epidemiology and the genetic biomarkers for BE and progression to dysplasia/EAC, and we feel that the manuscript has much improved.
Question 1. What is the prevalence of BE across Europe, USA and Asia and which are the possible causes of this variability?
Answer: The global prevalence of histologically confirmed BE in individuals with gastro-oesophageal reflux accoding to the Montreal definition [3] varies among countries, broadly resembling the geographical distribution of gastro-esophageal reflux itself. The highest prevalence is registered in Norh America (14.3%) and South America (12.5%), followed by Asia (6.1%), Europe (5.7%) and Middle-East (3.8%) [2]. This variability may be caused by different pathogenetic factors other than GERD itself and genetic characteristics [4]. First, the prevalence of Helicobacter pylori infection, which differs between the ethnic groups, has been found to be inversely associated with BE risk and may therefore act as a protective factor [5]. Second, obesity probably acts as an independent risk factor for BE, both through low levels of adiponectin and high levels of leptin that alter cell proliferation processes, and changes in the esophageal microbiota due to unhealthy dietary habits that promote carcinogenesis [6].
Question 2. Are there any biological biomarkers or genetic factors associated with BE and esophageal adenocarcinoma?
Answer: BE is a known precursor of EAC, which has led to the recommendation that patients with BE undergo regular endoscopic surveillance. However, the diagnostic yield of endoscopy with Seattle biopsy protocol has been questioned. Thus, in order to prevent the increasing morbidity and mortality from EAC, there is an urgent need for early detection and surveillance biomarker assays that are accurate, low-cost, and feasible to implement in clinical practice. Many biomarkers have been studied: altered TP53 tissue expression seems to be the most promising one [7], [8], whereas p16, and higher circulating levels of leptin, glucose, insulin, CRP, IL6, and sTNFR-2 may be also associated with an increased risk of EAC [9]. More prospective studies are required to identify biomarkers that can help select high-risk individuals for targeted prevention and early detection.
Reviewer 2 Report
Thanks for having me to review this interesting paper which I read with great interest. Overall this is a well-written paper addressing some important issues in the caring of BE patients. I have some comments and suggestions. 1) Question one: In reality some patients with typical GERD symptoms do have some other organic upper GI diseases, such as peptic ulcer or gastric or duodenal malignancy. Therefore indication of endoscopic examination depends on the clinical picture including, but not limited to, risk factors of BE. A more balanced statement may be needed here. 2) Question seven: A cost-effective study suggest an optimal age to stop surveillance of BE, but we should notice that health condition of patients and economic status of different countries and regions vary greatly. Some individualized approach may be required. 3) Line 132-133, it seems that some sentences are missed here.
Author Response
RESPONSE: We thank the reviewer for the insightful comments. We have rephrased the former question one (now question three) as follows: "Therefore, performing upper gastrointestinal (GI) endoscopy to rule out BE in all patients with chronic symptoms of GERD is not feasible; it is necessary to stratify the risk of BE and prioritize endoscopy in patients with a higher probability of having this condition. "
We have also modified the former question seven (now question 9) as follows: "Nevertheless, health condition of patients and economic status of different countries and regions vary greatly, therefore a tailored approach may be required."
We have also checked the typos.
Thank you again.
Reviewer 3 Report
The Authors presented a well-discussed opinion paper on Barrett'esophagus clincial management. The paper is well-written and presented and offers the readers a very good update on the state of the art on the subject. I only suggest to add a short paragraph on the molecular findings in BE and esophageal adenocarcinoma and possible subgrouping with clinical relevance (for an example see PMID 361396661).
Minor point:
phrase at line 133 seems to be incomplete. Please correct.
Author Response
We thank the reviewer for the appreciation of our work and the insightful comments. We have added a new paragraph (no.2) with molecular and genetic features of BE and progression risk towards EAC.
"Question 2. Are there any biological biomarkers or genetic factors associated with BE and esophageal adenocarcinoma?
Answer: BE is a known precursor of EAC, which has led to the recommendation that patients with BE undergo regular endoscopic surveillance. However, the diagnostic yield of endoscopy with Seattle biopsy protocol has been questioned. Thus, in order to prevent the increasing morbidity and mortality from EAC, there is an urgent need for early detection and surveillance biomarker assays that are accurate, low-cost, and feasible to implement in clinical practice. Many biomarkers have been studied: altered TP53 tissue expression seems to be the most promising one [7], [8], whereas p16, and higher circulating levels of leptin, glucose, insulin, CRP, IL6, and sTNFR-2 may be also associated with an increased risk of EAC [9]. More prospective studies are required to identify biomarkers that can help select high-risk individuals for targeted prevention and early detection."
Reviewer 4 Report
This is a good summary of clinicians' frequently asked questions about the management of patients with Barrett's esophagus. In my opinion, the material will be interesting for readers.
Author Response
Thank you for the appreciation of our work.
Reviewer 5 Report
Dear Editor,
Dear Authors,
I read with great interest the manuscript entitled “Management of Barrett’s esophagus: ten answers to ten questions” by Zagari RM et al. This was a well-written, brief opinion-review, addressing ten key topics in the management of BE. These topics are briefly discussed according to the most recent available literature and clearly explained to the readers. Thus, a practical guide for proper BE management, especially useful for non-expert in this field, is provided by the authors.
I have no comments and I consider the manuscript relevant for the research context.
Author Response
We thank you for appreciating our work.